# Magnetic bioassembly platforms for establishing craniofacial exocrine gland organoids as aging *in vitro* models

**Teerapat Rodboon**[1], **Glauco R. Souza**[2,3,4], **Apiwat Mutirangura**[5], **Joao N. Ferreira**[1]*

**1** Avatar Biotechnologies for Oral Health and Healthy Longevity Research Unit, Faculty of Dentistry, Chulalongkorn University, Bangkok, Thailand, **2** University of Texas Health Sciences Center at Houston, Houston, TX, United States of America, **3** Nano3D Biosciences Inc., Houston, TX, United States of America, **4** Greiner Bio-One North America Inc., Monroe, NC, United States of America, **5** Center of Excellence in Molecular Genetics of Cancer and Human Disease, Department of Anatomy, Faculty of Medicine, Chulalongkorn University, Bangkok, Thailand

* Joao.F@chula.ac.th

**Data Availability Statement:** All relevant data are within the paper and its Supporting Information files.

## Abstract

A multitude of aging-related factors and systemic conditions can cause lacrimal gland (LG) or salivary gland (SG) hypofunction leading to degenerative dry eye disease (DED) or dry mouth syndrome, respectively. Currently, there are no effective regenerative therapies that can fully reverse such gland hypofunction due to the lack of reproducible *in vitro* aging models or organoids required to develop novel treatments for multi-omic profiling. Previously, our research group successful developed three-dimensional (3D) bioassembly nanotechnologies towards the generation of functional exocrine gland organoids via magnetic 3D bioprinting platforms (M3DB). To meet the needs of our aging Asian societies, a next step was taken to design consistent M3DB protocols to engineer LG and SG organoid models with aging molecular and pathological features. Herein, a feasible step-by-step protocol was provided for producing both LG and SG organoids using M3DB platforms. Such protocol provided reproducible outcomes with final organoid products resembling LG or SG native parenchymal epithelial tissues. Both acinar and ductal epithelial compartments were prominent (21 ± 4.32% versus 42 ± 6.72%, respectively), and could be clearly identified in these organoids. Meanwhile, these can be further developed into aging signature models by inducing cellular senescence via chemical mutagenesis. The generation of senescence-like organoids will be our ultimate milestone aiming towards high throughput applications for drug screening and discovery, and for gene therapy investigations to reverse aging.

## 1. Introduction

Craniofacial exocrine glands, such as lacrimal glands (LG) and salivary glands, are essential organs that produce lubricating fluids from their acinar epithelia in the form of tears or saliva, respectively [1, 2]. In humans, LG acinar cells are serous-mucous but predominantly have mucous cells [2]. Meanwhile, humans have three major salivary glands—parotid, sublingual,

**Funding:** This research is funded by Thailand Science Research and Innovation Fund Chulalongkorn University (Grant number: CU_FRB65_hea (7)_013_32_08) to JNF and AM. This project is funded by the National Research Council of Thailand (NRCT), by a Mid-career Research Grant (Grant number: NRCT5-RSA63001-12) to JNF. Avatar Biotechnologies for Oral Health and Healthy Longevity Research Unit is funded by the Ratchadaphiseksomphot Endowment Fund, Chulalongkorn University (Grant number: 33/2565 : RU). TR is supported by a Postdoctoral Fellowship, Ratchadapisek Somphot Fund, Chulalongkorn University. The funders had no role in study design, data collection and analysis, decision to publish, or preparation of the manuscript.

**Competing interests:** The authors declare the following financial interests/personal relationships which may be considered as potential competing interests: Glauco R. Souza is employed by Greiner Bio-One International GmbH which produces NanoShuttle™ magnetic nanoparticles.

and submandibular glands (SMG)—but the latter are the relevant ones are mainly composed of mucous cells to provide the mucous secretion and oral moisture at rest [1]. Overall, epithelial secretory cells produce fluids that contain water, proteins, mucins, enzymes, and inorganic compounds to maintain a functional homeostasis in the ocular and oral cavities [3, 4]. Likewise, primary secretory fluids are synthesized by acinar epithelial units and transported to the external surfaces through an interconnected network of ducts, which is facilitated by the contractile action of myoepithelial cells [1, 2]. In addition to the functional and phenotypic similarities between LG and SMG, these two glands also share several clinical and pathological signatures.

Dry eyes and dry mouth syndromes are common disabling conditions among the elderly, resulting in epithelial dysfunction of the LG or SMG and a greatly reduced secretory fluids [5–7]. These syndromes lead to poor lubrication and moisture, which negatively affects routine daily activities (i.e. reading, speaking, chewing) and the quality of life of aging populations [5, 7]. In dry eyes syndrome (DES), long-term deficiency of tears may promote corneal epithelial damage and increase the risk of secondary infection. Also, painful inflammatory lesions in the oral mucosa linings occur in the oral cavity of patients with dry mouth syndrome (DMS) [6, 7]. DES and DMS involve cellular senescence-related factors due to biological aging; however, such can be aggravated by risk factors including polypharmacy in the elderly, autoimmunity, hormonal imbalances, radiotherapy modalities for head and neck cancers, among others [7–11]. Epidemiological studies clearly noted the high prevalence of both DES and DMS and its association with the aging phenomenon [11–13]. Hence, the age-related epithelial impairment of both craniofacial glands is a topic of interest for researchers and clinicians in the fields of dentistry as well as in head and neck pathology and oncology. Histological investigations on the aged human LG and SG confirmed that aging causes parenchymal acinar atrophy, which is associated with interstitial fibrosis and ductal hyperplasia [14, 15]. Though, preclinical translational models of LG/SG aging and effective treatment modalities to tackle it are lacking or scarce. Preclinical animal models for DED and DMS include rodents and swine [16–18]. However, phenotypic and functional observations indicate that rodent models have many limitations since they poorly represent pathophysiological mechanisms occurring in human craniofacial glands [17, 19–21]. Previously, anatomical and histological similarities have been reported between porcine and human LG and SG [22–24]. Also, the human resemblance of vascular and immune systems (as well as pathogenesis processes) with their porcine counterparts is remarkable and make porcine models suitable towards future clinical studies targeting DED or DMS therapies [22, 23, 25, 26]. Nonetheless, experimental research requires multiple levels of reproducibility and consistency to address pathogenesis, which cannot be provided by large scale *in vivo* animal models as these are time consuming, require substantial resources and do not favor 3R's principles in animal welfare (Replacement, Reduction and Refinement). Yet, the biofabrication of functionally competent LG and SG cultures *in vitro* or *ex vivo* is challenging since organoid protocols are lacking to maintain the multi-omic biological complexities of the native glands [27, 28]. To overcome this challenge, it is important to establish a consistent and reproducible *in vitro* organoid model to mimic epithelial cellular senescence and advance research towards an effective clinical management of DES and DMS. Previous murine studies have successfully shown the maintenance of epithelial progenitor and stem cell markers in two-dimensional (2D) LG and SG cell culture systems [29, 30]. However, these cells lack the ability to generate acinar and ductal compartments in 2D. Conversely, three-dimensional (3D) organoid platforms possess such ability to produce different epithelial compartments [28]. These 3D systems can support long-term cell viability, maintain stem/progenitor cell markers and potentially differentiate cells into mature epithelial organoids [28]. However, across most of the reported LG and SG organoid models, a large predominant ductal

compartment is produced, which functionally undermines the action of the very limited cluster of acinar secretory cells [28, 31].

Previously, our research group has established a successful strategy to assemble innervated functional epithelial SG organoids expressing acinar and ductal epithelial markers using a novel magnetic 3D bioassembly platforms with human and porcine primary cells [32, 33]. One of these nano-based platforms is named magnetic 3D bioprinting (M3DB) and can also be applied in the biofabrication of consistent and scalable LG organoids with high cell viability [24]. One of our research groups have also generated aging models using etoposide treatment to induce chemical mutagenesis and cellular senescence [34]. Herein, an optimized protocol is provided to develop an enriched acinar secretory LG/SG organoid with a ductal compartment, and amenable to cellular senescence induction towards future aging models. Such models will potentially enable novel gene therapies to reverse the aging phenomena in the LG and SG.

## 2. Material and methods

The protocol described in this peer-reviewed article is published on protocols.io. [https://dx. doi.org/10.17504/protocols.io.b5ttq6nn] and is included as a supporting information file with this article (S1 File).

## 3. Expected results

This protocol was developed to biofabricate LG or SG organoids that express parenchymal epithelial cell markers and can be used to investigate aging-related diseases in these glands. Further, this laboratory protocol can be divided into 3 steps as illustrated (Fig 1): 1) LG/SG cell isolation and epithelial cell differentiation; 2) organoid establishment; and 3) induction of cellular senescence in the organoid.

### 3.1 Primary cell isolation from porcine gland biopsies

This protocol was established for the LG and SG organoids. Although for a clear presentation of the preliminary data, LG organoid datasets are mainly displayed. Firstly, primary cells are isolated from LG/SG of a 3- to 5-month-old swine and an initial 2D monolayer culture is developed in expansion media (EM). To generate a LG with an aging signature, cells are cultured until reaching 70%-80% confluency, then such are sub-cultured for 3 passages while cell heterogeneity is still present (Fig 2). Within 4–6 culture days, epithelial clusters underwent growth and expansion, and 2 main phenotypes can be clearly observed: a large polygonal-like epithelial phenotype with predominant granular cytoplasm and a cell size diameter >20 μm (Fig 2), and a small polygonal-like epithelial phenotype with a limited cytoplasmic compartment and a cell size ≤20 μm (Fig 2). In addition, epithelial spherules were formed suggesting an ectodermal morphological origin often observed with human monolayer LG cells (Fig 2), as well as a dendritic cell population (Fig 2). However, these populations can be overtaken by fibroblast-like cells (Fig 2) after 3 passages (S1 Fig). To prevent this potential scenario, the monolayer culture system was enriched with epithelial-like cells by splitting the cells in EM for 2 days and then switch to a serum-free DKSFM supplemented with EGF, FGF-7 and FGF-10 for 7 days. Under this culture conditions, the numbers of epithelia-like cells are constantly increasing meanwhile the spindle-like cells are rapidly declining. Thus, we termed this media the "epithelial enrichment media" or EEM.

### 3.2 Epithelial profiling in 2D systems

Monolayer SG/LG cells were characterized by immunofluorescence assays against pro-acinar/acinar secretory (Aquaporin 5 or AQP5), myoepithelial/ductal progenitors (Cytokeratin 14, KRT14

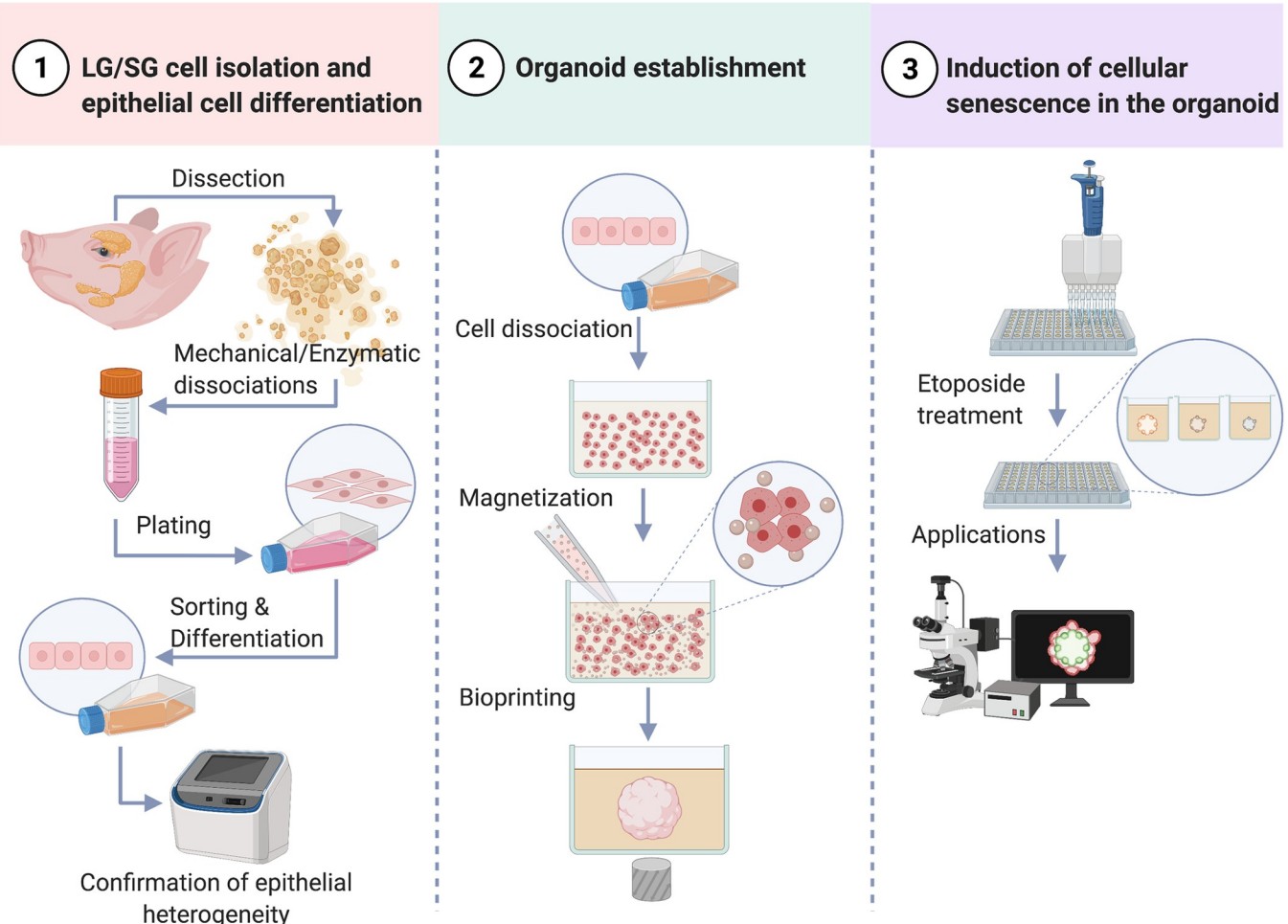

**Fig 1. Lab protocols for lacrimal gland (LG) and salivary gland (SG) organoid biofabrication via M3DB and induction of cellular senescence.** Created with BioRender.com.

or K14) and ductal epithelial markers (Cytokeratin 19, KRT19 or K19) (Fig 3) according to previous reports [31–33, 35, 36]. Based on their morphological features, most of AQP5 positive cells are small polygonal-like epithelial cells while the large polygonal-like epithelial cells mostly express ductal epithelial markers. Next, we investigated the number of epithelial cells after culture in EEM for 7 days by immunostaining the dissociated cells, and then quantifying such cell populations using a Countess 3 fluorescence automated cell counter. EEM-cultured LG cells retained the acinar (AQP5), myoepithelial/ductal progenitors (KRT14) and ductal epithelial populations (KRT19) predominantly (Fig 3). Cells expressed higher AQP5, KRT14, and KRT19 markers than in EM conditions (Fig 3), suggesting that EEM efficiently retained the acinar and ductal epithelial populations in 2D culture systems. Thus, the cell culture was designed to use epithelial-enriched 2D cells from passage 1 to passage 3 for further organoid biofabrication according to their morphological heterogeneity and population doubling time (S1 Fig).

### 3.3 LG organoid establishment

Next, the LG organoid was produced from the epithelial enriched LG cells by using our M3DB strategy. Herein, cells are dissociated and magnetized with a specific volume of magnetic

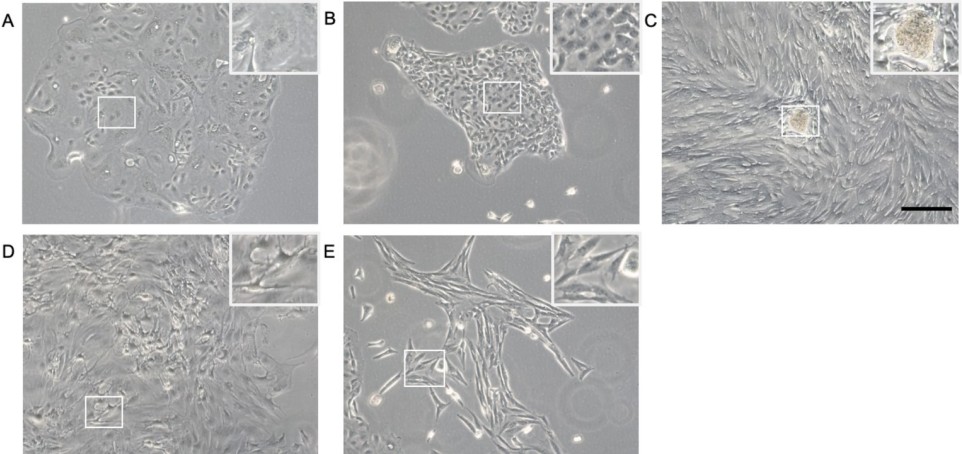

**Fig 2. Morphological heterogeneity of primary LG cells in monolayer cultures.** Primary cells isolated from porcine LG are cultured in expansion media for 7 days. The populations of large polygonal-like epithelial cells (A), small polygonal-like epithelial cells (B), epithelial spherule (C), dendritic cells (D), and fibroblast-like cells (E) are observed under phase-contrast microscopy at 20X of magnification. Scale bar: 200 μm.

nanoparticle solution (MNP) before assembling them into an organoid by using a magnetic spheroid drive (Fig 4 and S1 Movie). After organoids were cultured in EEM, there was a 5-fold increase in organoid size from 173 ±17.64 μm to 628 ±24.26 μm during 8 days of culture (Fig 4).

**3.3.1 Organoids displayed secretory and ductal epithelia.** Epithelial cell phenotype and polarization in LG organoids was assessed in the M3DB platform. Organoids exhibited acinar secretory epithelial cells (AQP5-positive) and also ductal epithelial cells (KRT19-positive) and myoepithelial/ductal progenitor cells (KRT14-positive) (Fig 5). Though, AQP5 was identified as a pro acinar marker in murine SG/LG, but the expression of such marker was showed in a population of cells on native SG of adult human [35, 36]. These cells were functionally responsive to parasympathetic stimulation with 10 μM of carbachol (Fig 5). In addition, to evaluate epithelial cell polarization in the organoids, the trans-epithelial electrical resistance (TEER) can be assessed after carbachol stimulation. The presence of a polarized epithelial compartment in M3DB-derived organoids can enhance the TEER (Fig 5). Overall, these findings indicate that the organoid have functional and polarized epithelial compartments in the LG organoid.

## 3.4 Induction of cellular senescence in organoids

To induce the cellular senescence in LG organoids, etoposide treatment (5–25 μM) was performed according to previous reports [34, 37, 38]. Next, cellular senescence in the organoids can be determined by measuring β-galactosidase activity, a known marker for senescent cells (Fig 6). Also, cellular senescence markers for genomic profiling include *P16*, *P21*, *Il6*, *Mcp1*, *Cxcl1*, and *Gdnf* and the replication independent endogenous DNA double strand breaks (RIND-EDSBs), which can be performed in the organoid platform (Fig 6) using the Minerva software suit after GeoMx Digital Spatial Profiling imaging (Nanostring, Seattle, WA, US), and transcriptome output plot from region of interest [39, 40]. Treatment with 10 μM etoposide generated a 50% reduction in cellular metabolism and was ideal to induce cellular senescence in the organoids without greatly compromising cell viability (Fig 6).

Overall, this protocol provides a feasible step-by-step comprehensive strategy to produce functional LG or SG organoids and their aging counterparts in the swine proof-of-concept

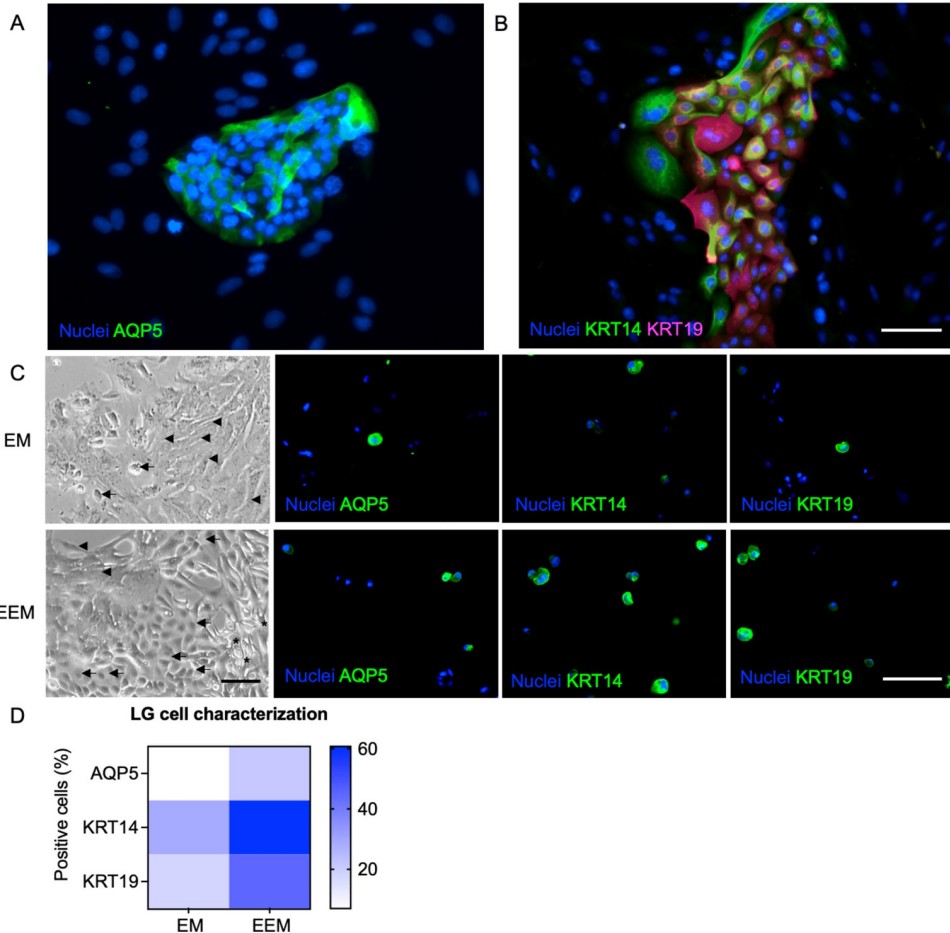

**Fig 3. Morphological and proteomic profiling of LG cells in 2D systems.** Acinar, ductal and myoepithelial/ductal progenitor compartments of the LG in 2D culture while in expansion media or EM (A). Differentiated cells after 7 days (C) showing numbers of epithelial-like cells (arrow), spindle shape cells (arrowhead) and dendritic cells (asterisk). Scale bar: 200 μm. AQP5, KRT14, and KRT19 protein markers were profiled and compared in both media conditions (EM versus EEM). Protein markers are quantified and displayed as a heat map with values as average % ±SE.

model. Furthermore, the organoids exhibited prominent acinar and ductal epithelial compartments.

## 4. Discussion

Aging involves a gradual and systemic impairment of organ and cellular physiology and has important repercussions in secretory epithelial functions of craniofacial exocrine glands leading to DES or DMS [11, 13, 15]. These pathological features are caused by genomic instability, a downstream pathway triggered by epigenetic modifications [41]. Previously, our team members reported a key epigenetic marker and offered the possibility to switch such key marker with gene therapy to reverse the aging process [42, 43]. Yet, certain challenges remain due to the lack of preclinical disease models to investigate such aging reversal process and its cellular senescence pathways and mechanisms. *In vivo* animal models can be timely aged, but this implies the consumption of several resources and time constraints. In the last decade, organoid models have offered relevant advantages in this regard as per comprehensive investigations done by Hans Clevers and his colleagues and deemed as feasible alternatives in line with the 3R's animal welfare principles.

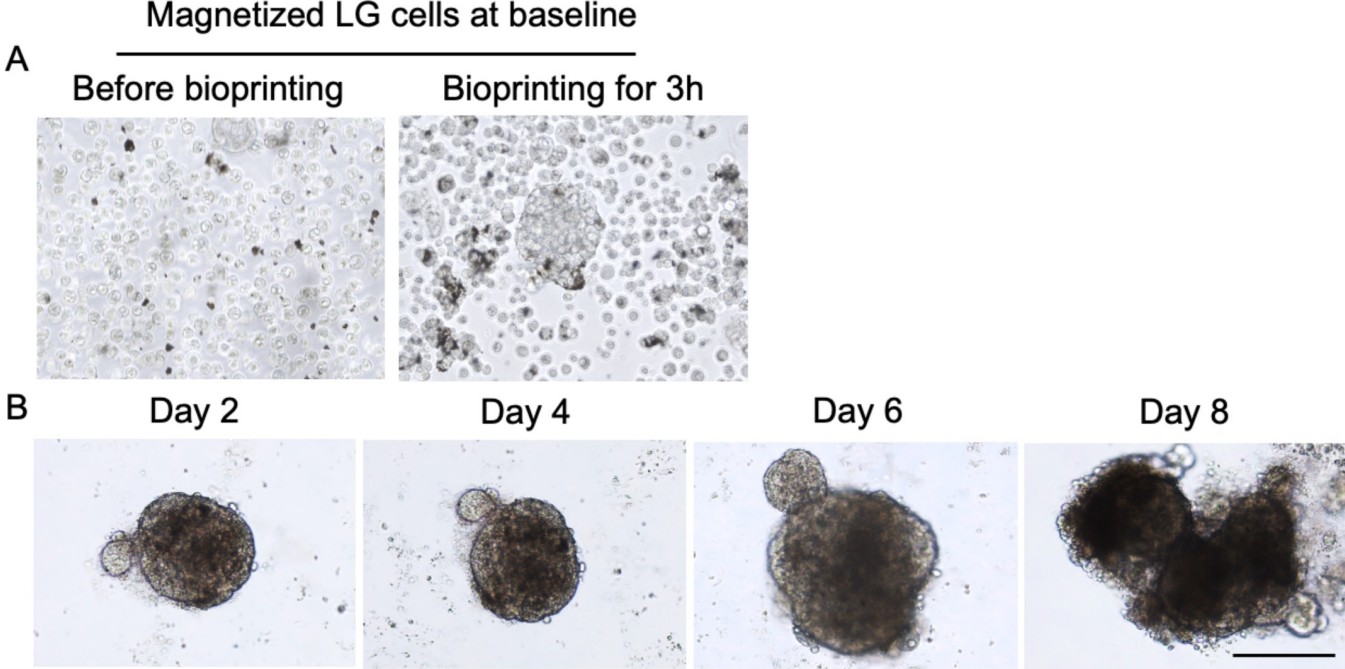

**Fig 4. Micrograph morphologies of cells or organoids from LG 2D culture system to organoid stages.** Differentiated LG cells in 2D were magnetized with magnetic nanoparticle solution (MNP) and then 3D assembled by M3DB (A). Organoids were cultured and their morphology was profiled via light microscopy and high throughput scanning analysis for 8 days. Scale bar: 200 μm.

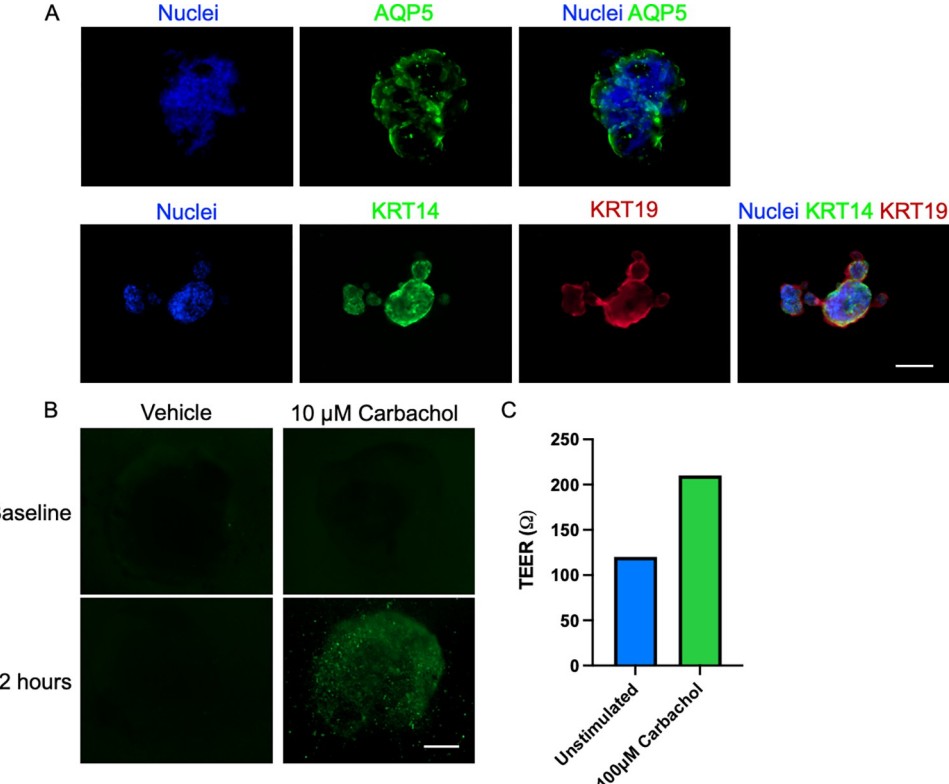

**Fig 5. Epithelial compartments and representation of functional datasets in LG organoids.** Expression of acinar, ductal and progenitor epithelial markers (A). Epithelial function evaluated by $Ca^{2+}$ uptake assays after parasympathetic stimulation with carbachol (B) and trans-epithelial electrical resistance (TEER) (C) in LG organoids. Scale bar: 200 μm.

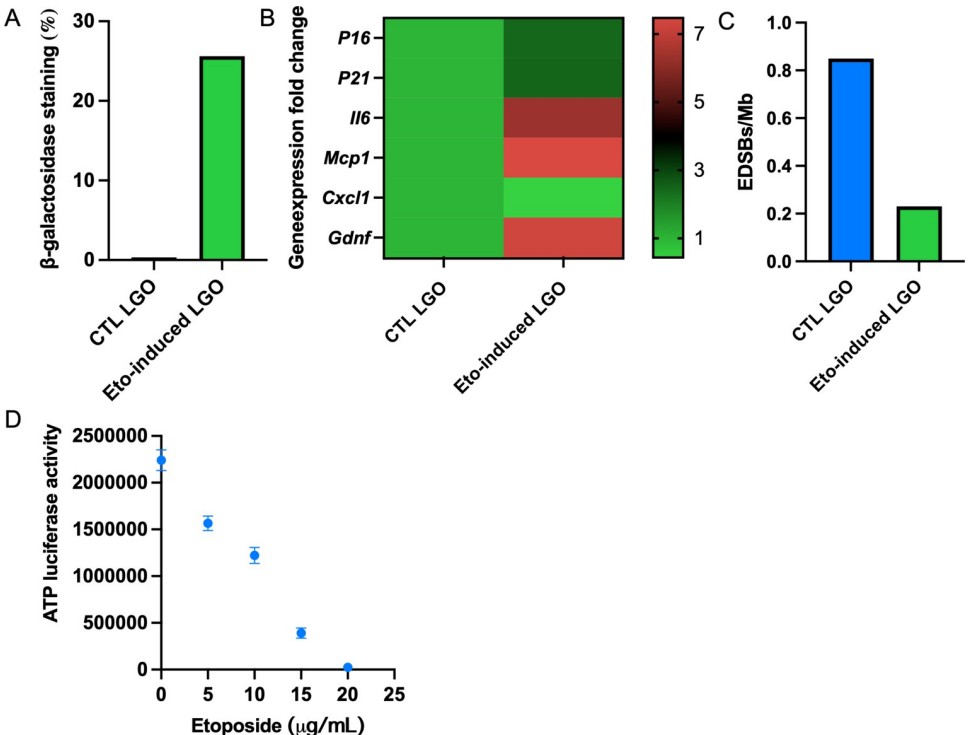

**Fig 6. Multi-omics profiling of cellular senescence-induced LG organoids.** Senescence markers are evaluated including β-galactosidase activity (A), gene expression arrays (B), and levels of RIND-EDSBs (C) after inducing the organoid with etoposide. For inducing cellular senescence in the organoid, the concentration of etoposide required to create a 50% reduction in metabolism was 10 μM as determined by ATP-luciferase activity with CellTiter-Glo® 3D kit from Promega, USA (D). Abbreviations: CTL LGO–control LG organoid with normal epithelial function; Eto-induced LGO–etoposide-induced LG organoid with impaired epithelial function.

Previously, our research groups have successfully generated reproducible and functional epithelial LG- or SG-like organoids via M3DB platforms [24, 32]. Herein, a protocol is proposed for creating preclinical disease models with aging multi-omic signatures for LG/SG organoids. As part of this novel biofabrication strategy, we established 3D organoids from epithelial enriched cells in 2D culture systems and perform cell sorting accordingly to the epithelial compartment that we need. Primary cells provided a phenotypic heterogeneity until passage 3 and this is a hallmark of human LG cells alike previous report [44]. More importantly, organoids displayed functional acinar and ductal compartments together with epithelial progenitors, in response to parasympathetic stimulation. Thus, this bio-printed exocrine gland organoid platform can be utilized as an avatar model with an aging signature and cellular senescence features resembling those observed in DES and DMS. These aging LG/SG models constitute a unique opportunity to investigate the senescence multi-omic markers such as β-galactosidase, *p16*, *p21*, *Il6*, *Mcp1*, *CxCl1*, and *Gdnf* at genomic, proteomic, and even mitochondrial levels using spatial biology imaging strategies. Spatial biology profiling approaches have recently been used to generate publicly available resources such as online organ atlas [40, 45]. These resources allow researchers to unveil the molecular, physiologic, and pathological mechanisms in human epithelial organs though only limited to the pancreas, colon and kidney. In addition, only human and mouse spatial organ atlas exist, porcine multi-omics panels (for transcriptome and proteome) have not been validated. Hence, the validation of porcine high-plex spatial molecular imaging platforms is a key step towards the establishment of swine preclinical models.

Regarding aging models, a novel senescence marker called RIND-EDSBs has been proposed by one of our research groups led by Mutirangura and colleagues [43]. These endogenous DNA double-strand breaks are enriched in the methylated heterochromatic areas of the human genome and can be repaired by ATM-dependent non-homologous end-joining pathway. As part of our ongoing work, these pathways are currently being targeted to switch or reverse the aging phenomena by gene therapy strategies focused on halting the genomic instability and cellular senescence.

## Supporting information

**S1 File. Step-by-step laboratory protocol.** This protocol was developed at protocols.io, which can be assessed via this DOI: [https://dx.doi.org/10.17504/protocols.io.b5ttq6nn].
(PDF)

**S1 Movie. Bioprinting of magnetized LG cells via M3DB.** Cells were magnetized and bioprinted over a magnetic dots located under the 96-well plate.
(MOV)

**S1 Fig. LG cell morphology in expansion media.** The morphology of LG cells in expansion media (EM) and epithelial enrichment media (EEM) up to passage 4.
(TIF)

## Acknowledgments

Authors would like to acknowledge the following individuals: Dr. Narumol Bhummaphan from Department of Anatomy, Faculty of Medicine at Chulalongkorn University (Thailand) for experimental support and training towards the implementation of cellular senescence techniques; Dr. Toan Van Phan and Ms. Yamin Oo for their support on the cell culture protocol during organoid development; and Dr. Jirawat Chuaykaew, a pathologist from the Department of Anatomical Pathology, Chonburi Hospital (Thailand) for training and consultation on anatomical, morphological and pathological analysis.

## Author Contributions

**Conceptualization:** Glauco R. Souza, Apiwat Mutirangura, Joao N. Ferreira.

**Data curation:** Teerapat Rodboon.

**Formal analysis:** Teerapat Rodboon, Joao N. Ferreira.

**Funding acquisition:** Joao N. Ferreira.

**Investigation:** Teerapat Rodboon.

**Methodology:** Teerapat Rodboon.

**Project administration:** Joao N. Ferreira.

**Resources:** Glauco R. Souza, Apiwat Mutirangura, Joao N. Ferreira.

**Software:** Joao N. Ferreira.

**Supervision:** Apiwat Mutirangura, Joao N. Ferreira.

**Validation:** Teerapat Rodboon.

**Visualization:** Glauco R. Souza, Apiwat Mutirangura.

**Writing – original draft:** Teerapat Rodboon.

**Writing – review & editing:** Glauco R. Souza, Apiwat Mutirangura, Joao N. Ferreira.

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
