## [Decision Letter · Decision Letter 0]

10 May 2022

PONE-D-22-06068Establishment of craniofacial exocrine gland organoid magnetic bioassembly platforms as aging multi-omic signaturesPLOS ONE

Dear Dr. Ferreira,

Thank you for submitting your manuscript to PLOS ONE. After careful consideration, we feel that it has merit but does not fully meet PLOS ONE’s publication criteria as it currently stands. Therefore, we invite you to submit a revised version of the manuscript that addresses the points raised during the review process.

Please also give the response to the reviewers' comments point by point.

We look forward to receiving your revised manuscript.

Kind regards,

Li-Ping Liu

Academic Editor

PLOS ONE

Journal Requirements:

2. Please upload a copy of Figure 6, to which you refer in your text on page 18. If the figure is no longer to be included as part of the submission please remove all reference to it within the text.

Reviewers' comments:

Reviewer's Responses to Questions

**Comments to the Author**

1. Does the manuscript report a protocol which is of utility to the research community and adds value to the published literature?

Reviewer #1: Yes

Reviewer #2: Yes

Reviewer #3: Yes

2. Has the protocol been described in sufficient detail?

Descriptions of methods and reagents contained in the step-by-step protocol should be reported in sufficient detail for another researcher to reproduce all experiments and analyses. The protocol should describe the appropriate controls, sample sizes and replication needed to ensure that the data are robust and reproducible.

Reviewer #1: Partly

Reviewer #2: Yes

Reviewer #3: Partly

3. Does the protocol describe a validated method?

Reviewer #1: Yes

Reviewer #2: No

Reviewer #3: Yes

4. If the manuscript contains new data, have the authors made this data fully available?

Reviewer #1: Yes

Reviewer #2: No

Reviewer #3: N/A

**5. Is the article presented in an intelligible fashion and written in standard English?**

Reviewer #1: Yes

Reviewer #2: **No: **Non-standard English is sometimes used.

Reviewer #3: Yes

6. Review Comments to the Author

Reviewer #1: This paper is definitely valuable for the researchers concerned to this topic. however, following minor changes are advised.

1) The abstract needs to be revised addressing directly to the targeted problem (dry eyes and dry mouth syndrome) as it might give readers a clear idea about the focus of this protocol.

2)The figures need to be improved quality wise.

Reviewer #2: The authors used previously published techniques to create 3D porcine LG spheroids including magnetic bioprinting for spheroid production. Some of the previous work used SG and the authors state the methods herein are valid for lacrimal gland (LG) and salivary gland (SG), but they only show LG data in the paper. They demonstrate scant levels of the water channel, Aqp5, which is used as a marker for proacinar/acinar cells. Aqp5 is not a mature acinar marker, as stated in the text, and no mature acinar markers are shown. They show higher levels of ductal cytokeratins in both EM and EEM, although EEM does seem to enrich for epithelium relative to unspecified cellular composition. They then use an unspecified treatment with etoposide to induce cellular senescence in the spheroids. The primary data and methods are not specified but they show upregulation of senescence markers with treatment. They claim these data establish an exocrine gland senescence in vitro model for drug discovery to combat aging-related loss of exocrine gland function. However, showing that DNA damage induced with etoposide treatment of poorly differentiated spheroids in vitro does not at all demonstrate an in vitro model of drug discovery to combat aging related loss of exocrine gland function. Although a protocol is provided for making the porcine organoids, albeit not in the same level of detail, this method has been previously published. Overall, the data are poorly described, of poor quality with no mention of n or statistical analysis, incomplete, and they do not support the claims.

Other specific issues:

The authors show data in Figure 6 which represents an unspecified method of multiomics profiling. The figure is not explained or referred to in the text.

They refer to spatial profiling in the discussion and show a screen shot from an unrelated figure as a supplementary figure. This should be removed.

The title is very confusing

Non-standard English is sometimes used (i.e. salivary submandibular glands).

Reviewer #3: Major comments

1. The salivary gland and especially the lacrimal gland express AQP5 in acinar cells and intercalated ductal cells. In addition, AQP5 is one of the pro-acinar markers. Thus, it is suggested to stain with bHLHA15 (Mist1) is better to detect mature acinar cells in SG and LG.

2. Similar to acinar's pro/mature differentiation, mature luminal ductal cells in SG and LG have expressed cytokeratin 7 with cytokeratin 19. Developing luminal ductal cells or luminal ductal progenitor cells express cytokeratin 19, not cytokeratin 7. Therefore, if you want to express 'mature,' you should stain using cytokeratin 7 antibodies, not cytokeratin 19 antibodies.

3. The shape and size of cells often show important features of specialized cells. Acinar cells are usually 3 times bigger than ductal cells. However, the results oppositely express cellular markers. Thus, it would be better to precisely compare the character of small-polygonal and large-polygonal cells using various acinar cell-related genes and ductal cell-related genes.

Minor comments

1. The authors wrote that 100 mM carbachol was used in experiments. However, carbachol and its derivatives are difficult to dissolve in solvents (water, DMSO, or etc.) over 100 mM. Please make sure that you are trying to write 100 μM.

2. The authors said that EMM constantly grows epithelial-like cells; however, the result was not. Replace the results of passage 4 with one that matches your description, or explain what happens from that point with the population doubling graph.

3. Epithelial spherules can often be identified in the 2D culture of cells derived from organs of ectodermal origin. It is not a particular phenomenon seen only in Human LG cell culture. Please correct the expression.

7. PLOS authors have the option to publish the peer review history of their article (what does this mean?). If published, this will include your full peer review and any attached files.

Reviewer #1: No

Reviewer #2: No

Reviewer #3: No

---

## [Author Response · Author response to Decision Letter 0]

23 Jun 2022

2) REVIEWER 1:

2.1) This paper is definitely valuable for the researchers concerned to this topic. however, following minor changes are advised.

Authors response: Thank you for your input. We have revised the manuscript according to your suggestions.

2.2) The abstract needs to be revised addressing directly to the targeted problem (dry eyes and dry mouth syndrome) as it might give readers a clear idea about the focus of this protocol.

Authors response: A clarification was indeed necessary, and hence, we rewrote the entire abstract of the revised manuscript (page 1, lines 2-23) to show a clear and better focus. 

2.3) The figures need to be improved quality wise.

Authors response: We agree this improvement is necessary since the PLOS submission platform seems to decrease the resolution of our 300dpi images. We have improved the resolution of all figures and re-uploaded those figures into the PLOS platform as 600dpi.

3) REVIEWER 2:

3.1) The authors used previously published techniques to create 3D porcine LG spheroids including magnetic bioprinting for spheroid production. Some of the previous work used SG and the authors state the methods herein are valid for lacrimal gland (LG) and salivary gland (SG), but they only show LG data in the paper. 

Authors response: Thank you for your input. Allow us to clarify that this manuscript is targeting a "Lab Protocol" publication type in PLOS ONE and not a regular original research article. The requirements for such "Lab Protocol" section are listed here: https://plos.org/open-science/open-methods

Therefore, according to PLOS ONE requirements for a "Lab Protocol", we are showing a step-by-step protocol with our bioprinting strategy that is based on what we had previously reported with primary cells from the LG and SG, which have similar outcomes: 

DOI: https://doi.org/10.1016/j.slasd.2021.11.002

DOI: https://doi.org/10.1002/term.2809

In summary, we are offering to the readership this step-by-step protocol to produce exocrine gland organoids from primary cell cultures from porcine LG and SG. 

3.2) They demonstrate scant levels of the water channel, Aqp5, which is used as a marker for proacinar/acinar cells. Aqp5 is not a mature acinar marker, as stated in the text, and no mature acinar markers are shown. They show higher levels of ductal cytokeratins in both EM and EEM, although EEM does seem to enrich for epithelium relative to unspecified cellular composition. 

Authors response: After our extensive IHC/ICC experimental trials, we could not validate specific antibodies against bHLHA15 (Mist1) or amylase marker in the porcine SG/LG (Sus scofra domesticus), which are commercially available for other species. We could only use the anti-AQP5 antibody for the characterization of "pro-acinar" cells as stated by the reviewer. The majority of the published reports have looked at mouse SG/LG and identified APQ5 as a "pro-acinar" marker. The SG mouse molecular anatomy atlas made available by the NIDCR/NIH (https://sgmap.nidcr.nih.gov/sgmap/sgexp.html) also confirms it. However, in light of recent reports this "pro-acinar" tag is debatable for the human or porcine SG/LG. In fact, the use of AQP5 antibody can support the identification of mature acinar cells in the human SG organ as revealed by a recent study published in the Journal of Dental Research by Drs. Wu's and Farach-Carson's group: https://doi.org/10.1177/00220345221076122 . In this study, this SG research group clearly showed the clear and prominent expression of AQP5 in the mature MIST1+ acinar SG compartment in the native human SG. Another earlier study has also showed the wide expression of AQP5 in the mature acinar parenchyma expressing alpha-amylase in the native human parotid gland: https://doi.org/10.1089/ten.tea.2016.0466

Due to the debatable nature of the AQP5 marker, we removed the word “mature” from the manuscript text on sections 3.2-3.3.1 (pages 8-9) and inserted references on page 8 to support the claim that AQP5 can be considered a secretory acinar marker in humans/pigs. 

3.3) They then use an unspecified treatment with etoposide to induce cellular senescence in the spheroids. The primary data and methods are not specified but they show upregulation of senescence markers with treatment. 

Authors response: To clarify this, we have added more information and data on etoposide treatment in the Introduction section (page 5, line 17-19) and Results section 3.2 (page 9, line 2-13). To support further this claim, we added references and also etoposide data on Figure 6 panel. The induction of cellular senescence and the aging phenomena by etoposide has been extensively described in the literature by our co-authors (DOI: 10.1096/fba.2021-00131) and other researchers:

DOI: https://doi.org/10.3390/cells10061466

DOI: https://doi.org/10.1038/oncsis.2015.37

We have inserted these 3 references on page 9 to specifically support the claim that cellular senescence can be induced by etoposide via chemical mutagenesis.

3.4) They claim these data establish an exocrine gland senescence in vitro model for drug discovery to combat aging-related loss of exocrine gland function. However, showing that DNA damage induced with etoposide treatment of poorly differentiated spheroids in vitro does not at all demonstrate an in vitro model of drug discovery to combat aging related loss of exocrine gland function.

Authors response: In this "Lab protocol" manuscript type, we showed that the differentiated organoids produced by our bioprinting protocol can respond to neurostimulation. Also, such organoids can be induced to a senescent state by treatment with etoposide at 10 �M. As mentioned on our previous response in 3.3): To support further this claim, we added information and references (see page 9, line 2-13) and also data on Figure 6 panel. The induction of cellular senescence and the aging phenomena by etoposide has been extensively described in the literature by our co-authors (DOI: 10.1096/fba.2021-00131 and other researchers:

DOI: https://doi.org/10.3390/cells10061466

DOI: https://doi.org/10.1038/oncsis.2015.37

 In addition, our 3 previous published reports and our functional data presented herein support that the “poorly differentiated spheroids” (stated by the reviewer) are rather composed of mature and functional epithelial compartments since the proposed acinar and ductal markers used have been validated in human SG studies published recently in the Journal of Dental Research (https://doi.org/10.1177/00220345221076122). 

We have proposed in this manuscript to use this platform for high throughput applications towards drug screening and gene therapy to reverse aging. Such gene therapy work is ongoing and is not the focus of this specific “Lab protocol”. We have clarified this on page 11, line 12-15: “As part of our ongoing work, these pathways are currently being targeted to switch or reverse the aging phenomena by gene therapy strategies focused on halting the genomic instability and cellular senescence.”

3.5) Although a protocol is provided for making the porcine organoids, albeit not in the same level of detail, this method has been previously published. Overall, the data are poorly described, of poor quality with no mention of n or statistical analysis, incomplete, and they do not support the claims.

Authors response: As we have stated earlier, this manuscript is targeting a "Lab Protocol" publication type in PLOS ONE. Therefore, we are showing a step-by-step protocol with our bioprinting strategy that is based on what we had previously reported with primary cells from the LG and SG. We have provided with our manuscript submission a comprehensive step-by-step protocol create at the protocols.io with the DOI link: https://dx.doi.org/10.17504/protocols.io.b5ttq6nn

3.6) Other specific issues: The authors show data in Figure 6 which represents an unspecified method of multiomics profiling. The figure is not explained or referred to in the text.

Authors response: This Figure 6 is now referred in the revised manuscript text in page 9, lines 5, 8 and 13. To offer more information on this method, a more extended explanation is provided now on page 9, line 2-13, as well as in the caption of this Figure 6 on page 15.

3.7) They refer to spatial profiling in the discussion and show a screen shot from an unrelated figure as a supplementary figure. This should be removed.

Authors response: We agree this might be confusing. Therefore, this supplementary figure was completely removed and such spatial profiling information can be assessed now through a data repository: https://osf.io/582rh/?view_only=f2dbb951b56b4429b29ba6d7b7ba7061

3.8) The title is very confusing

Authors response: We have changed the manuscript title to: "Magnetic bioassembly platforms for establishing craniofacial exocrine gland organoids as aging in vitro models". We hope this revised title can meet the expectations. 

3.9) Non-standard English is sometimes used (i.e. salivary submandibular glands).

Authors response: The manuscript was revised by an English native speaker, and corrections to scientific terminology were made through the entire manuscript.

4) REVIEWER 3:

4.1) The salivary gland and especially the lacrimal gland express AQP5 in acinar cells and intercalated ductal cells. In addition, AQP5 is one of the pro-acinar markers. Thus, it is suggested to stain with bHLHA15 (Mist1) is better to detect mature acinar cells in SG and LG.

 Authors response: In this study, we used primary cells from porcine LG/SG as a proof-of-concept model according to their similarities to human rather than to the mouse counterparts. We showed such similarities between humans and porcine in our previous report: (https://slas-discovery.org/action/showPdf?pii=S2472-5552%2821%2900017-4). Yet, after our IHC/ICC experimental trials, we could not validate specific antibodies against bHLHA15 (Mist1) markers in the porcine (Sus scofra domesticus), which are commercially available for other species. Hence, we could only use the anti-AQP5 antibody for the characterization of "pro-acinar" cells. However, this "pro-acinar" name tag is debatable: more recent literature using the AQP5 antibody can support the identification of mature acinar cells in the human SG as revealed by a recent study published in the Journal of Dental Research by Drs. Wu's and Farach-Carson's group: https://doi.org/10.1177/00220345221076122 . These authors clearly showed the marked expression of AQP5 in mature MIST1+ acinar SG compartments. In addition, another earlier study done in the native human adult parotid gland has also showed the wide expression of AQP5 in the mature acinar parenchyma expressing alpha-amylase: https://doi.org/10.1089/ten.tea.2016.0466

Due to the debatable nature of the AQP5 marker, we removed the word “mature” from the manuscript text on sections 3.2 and 3.3.1 (pages 8-9) and inserted references on page 8 to support the claim that AQP5 can be considered a secretory acinar marker in human/pig SG/LG tissues. 

4.2) Similar to acinar's pro/mature differentiation, mature luminal ductal cells in SG and LG have expressed cytokeratin 7 with cytokeratin 19. Developing luminal ductal cells or luminal ductal progenitor cells express cytokeratin 19, not cytokeratin 7. Therefore, if you want to express 'mature,' you should stain using cytokeratin 7 antibodies, not cytokeratin 19 antibodies.

Authors response: According to our extensive antibody validation experiments, only antibodies against two specific protein markers - cytokeratin 14 and cytokeratin 19 - can be used towards the characterization of the ductal compartment in porcine gland tissues. Our immunocytochemistry and IHC showed immunoreactivity for both cytokeratin 14 and cytokeratin 19 markers, which are increased in EEM treatment. The use of cytokeratin 19 is supported by a comprehensive study published by Drs. Wu's and Farach-Carson's group, which validated the use of cytokeratin 19 as a specific protein marker to identify mature ductal cells in the human SG: https://doi.org/10.1177/00220345221076122

Due to the debatable nature of the cytokeratin 19 marker, we removed the word “mature” from the manuscript text on sections 3.2 and 3.3.1 (pages 8-9) and inserted references on page 8 to support the claim that cytokeratin 19 can be considered a differentiated ductal marker in human/pig SG tissues. 

4.3) The shape and size of cells often show important features of specialized cells. Acinar cells are usually 3 times bigger than ductal cells. However, the results oppositely express cellular markers. Thus, it would be better to precisely compare the character of small-polygonal and large-polygonal cells using various acinar cell-related genes and ductal cell-related genes.

Authors response: We concur and look forward to clarify this. Herein, we can only state that our 2D culturing protocol provided most of the AQP5 positive acinar cells, which are small polygonal-like epithelial cells in the immunocytochemistry; while the large polygonal-like epithelial cells mainly express ductal progenitors and ductal mature epithelial markers (cytokeratin 14 and cytokeratin 19, respectively). To further confirm these findings, a single cell analysis is more relevant and will be performed in the future since this is mainly a lab protocol based on previous supporting data. The challenge with gene expression arrays of cytokeratins (cytokeratin 14 in particular) in the SG is that their expression levels do not match their respective protein expression as one can observe in the SG molecular anatomy atlas made available by the NIDCR/NIH (https://sgmap.nidcr.nih.gov/sgmap/sgexp.html). In addition, our 2D cell morphological observations are supported by our previous publication (DOI: https://doi.org/10.1016/j.slasd.2021.11.002).

4.4) Minor comments: The authors wrote that 100 mM carbachol was used in experiments. However, carbachol and its derivatives are difficult to dissolve in solvents (water, DMSO, or etc.) over 100 mM. Please make sure that you are trying to write 100 μM.

Authors response: We regret for the confusion caused. We clarified and replaced "100 µM" carbachol with "10 µM" throughout the manuscript.

4.5) Minor comments: The authors said that EMM constantly grows epithelial-like cells; however, the result was not. Replace the results of passage 4 with one that matches your description, or explain what happens from that point with the population doubling graph.

Authors response: We have clarified such concern and rewrote the outcomes on the revised manuscript on page 7, lines 21-24: "Thus, the cell culture was designed to use epithelial-enriched 2D cells from passage 1 to passage 3 for further organoid biofabrication according to their morphological heterogeneity and population doubling time (Fig S1)."

4.6) Minor comments: Epithelial spherules can often be identified in the 2D culture of cells derived from organs of ectodermal origin. It is not a particular phenomenon seen only in Human LG cell culture. Please correct the expression.

Authors response: We agree with such and rewrote the sentence on page 6 lines 23-24: "In addition, epithelial spherules were formed suggesting an ectodermal morphological origin often observed with human monolayer LG cells (Fig 2), as well as ...." 

We hope we have clarified all raised concerns in this newly revised version. Thank you again for your time and for considering this revised manuscript.

---

## [Decision Letter · Decision Letter 1]

20 Jul 2022

PONE-D-22-06068R1Magnetic bioassembly platforms for establishing craniofacial exocrine gland organoids as aging in vitro modelsPLOS ONE

Dear Dr. Ferreira,

Thank you for submitting your manuscript to PLOS ONE. After careful consideration, we feel that it has merit but does not fully meet PLOS ONE’s publication criteria as it currently stands. Therefore, we invite you to submit a revised version of the manuscript that addresses the points raised during the review process.

We look forward to receiving your revised manuscript.

Kind regards,

Li-Ping Liu

Academic Editor

PLOS ONE

Journal Requirements:

Reviewers' comments:

Reviewer's Responses to Questions

**Comments to the Author**

1. Does the manuscript report a protocol which is of utility to the research community and adds value to the published literature?

Reviewer #1: Yes

Reviewer #2: Yes

Reviewer #3: Yes

2. Has the protocol been described in sufficient detail?

Descriptions of methods and reagents contained in the step-by-step protocol should be reported in sufficient detail for another researcher to reproduce all experiments and analyses. The protocol should describe the appropriate controls, sample sizes and replication needed to ensure that the data are robust and reproducible.

Reviewer #1: Yes

Reviewer #2: Yes

Reviewer #3: Partly

3. Does the protocol describe a validated method?

Reviewer #1: Yes

Reviewer #2: Yes

Reviewer #3: Yes

4. If the manuscript contains new data, have the authors made this data fully available?

Reviewer #1: Yes

Reviewer #2: Yes

Reviewer #3: Yes

**5. Is the article presented in an intelligible fashion and written in standard English?**

Reviewer #1: Yes

Reviewer #2: Yes

Reviewer #3: Yes

6. Review Comments to the Author

Reviewer #1: I thank the authors of the article for fulfilling the last minor issues. I accept this article for publication.

Reviewer #2: Revisions to the figures are still required. Carbocol is still shown in figure 5 as 100 micromolar, even though the authors stated it is an error. Supplementary figure 2 showing spatial profiling in lung cancer still remains.

On the issue of differentiation, if all instances of "mature" differentiation or implication that the organoids are mature or fully differentiated are removed, then the AQP5 staining is fine and is informative as to the cell lineage. In the mouse, AQP5 is expressed in both embryonic "proacinar" cells as well as in adult acinar cells and so showing only AQP5 is insufficient to make a statement regarding maturity. There are insufficient studies in porcine tissue to know if its expression is similar.

Reviewer #3: The authors tried to meet the referees' queries sufficiently, and this revised article is worth publishing.

7. PLOS authors have the option to publish the peer review history of their article (what does this mean?). If published, this will include your full peer review and any attached files.

Reviewer #1: No

Reviewer #2: No

Reviewer #3: No

---

## [Author Response · Author response to Decision Letter 1]

20 Jul 2022

RESPONSE TO REVIEWER COMMENTS

1) REVIEWER #1:

1.1) I thank the authors of the article for fulfilling the last minor issues. I accept this article for publication.

Authors response: Thank you for your input and for the time taken to review this manuscript. 

2) REVIEWER #2:

2.1) Revisions to the figures are still required. Carbocol is still shown in figure 5 as 100 micromolar, even though the authors stated it is an error. Supplementary figure 2 showing spatial profiling in lung cancer still remains.

Authors response: Thank you for your feedback. We regret for the confusion caused. We may have accidentally uploaded the old figures (Fig 5 and S2 Figure were not correct). Please see the newly revised Figure 5 where we replaced "100 µM" carbachol with "10 µM". We also found one instance where carbachol concentration had a typo and we corrected it on the R2 revised version (see page 8 line 17). We also completely removed Supplementary Figure 2 from the submission platform as it was uploaded by mistake.

2.2) On the issue of differentiation, if all instances of "mature" differentiation or implication that the organoids are mature or fully differentiated are removed, then the AQP5 staining is fine and is informative as to the cell lineage. In the mouse, AQP5 is expressed in both embryonic "proacinar" cells as well as in adult acinar cells and so showing only AQP5 is insufficient to make a statement regarding maturity. There are insufficient studies in porcine tissue to know if its expression is similar.

Authors response: We agree with the reviewer's statement regarding the mouse AQP5 expression and the fact that there are insufficient studies for AQP5 in porcine salivary and lacrimal gland tissue. In this revised version, we did make sure that all instances of "mature" or "full" differentiation or the implications for such were completely removed (see page 9 line 15-16 and page 10 line 16-17). 

3) REVIEWER #3:

3.1) The authors tried to meet the referees' queries sufficiently, and this revised article is worth publishing.

Authors response: Thank you for your feedback and for the time taken to review this manuscript. 

We hope we have clarified the raised minor concerns with this newly revised version R2. Thank you again for your time and for considering this revised manuscript.

Sincerely,

João N. Ferreira, DDS MS PhD (Corresponding author on behalf of all authors)

E-mails: Joao.F@chula.ac.th

---

## [Editor Report · Decision Letter 2]

25 Jul 2022

Magnetic bioassembly platforms for establishing craniofacial exocrine gland organoids as aging in vitro models

PONE-D-22-06068R2

Dear Dr. Ferreira,

We’re pleased to inform you that your manuscript has been judged scientifically suitable for publication and will be formally accepted for publication once it meets all outstanding technical requirements.

Kind regards,

Li-Ping Liu

Academic Editor

PLOS ONE
---

## [Editor Report · Acceptance letter]

27 Jul 2022

PONE-D-22-06068R2 

Magnetic bioassembly platforms for establishing craniofacial exocrine gland organoids as aging *in vitro* models 

Dear Dr. Ferreira:

I'm pleased to inform you that your manuscript has been deemed suitable for publication in PLOS ONE. Congratulations! Your manuscript is now with our production department. 

Kind regards, 

on behalf of

Dr. Li-Ping Liu 

Academic Editor

PLOS ONE